# How Effective Are Time Series Foundation Models for Epidemiological Data Analysis?

Lucas Moreira Ribeiro
*Postgraduate Program in Computer Science*
*Federal University of Ouro Preto*
Ouro Preto, MG, Brazil
lucas.mr@aluno.ufop.edu.br

Eduardo Jose da Silva Luz
*Department of Computing*
*Federal University of Ouro Preto*
Ouro Preto, MG, Brazil
eduluz@ufop.edu.br

Jadson Castro Gertrudes
*Department of Computing*
*Federal University of Ouro Preto*
Ouro Preto, MG, Brazil
jadson.castro@ufop.edu.br

*Abstract*—Understanding the capabilities and limitations of modern time series models is essential for their effective use in domain-specific contexts. This study investigates the zero-shot forecasting performance of state-of-the-art Time Series Foundation Models (TSFMs) using real-world epidemiological data. Guided by a central hypothesis and three research questions, we benchmark several TSFMs, including TimesFM, ChronosT5Base, and Moirai variants, on monthly time series of notifiable diseases in Brazil, considering different forecast horizons and context lengths. Through robust evaluation using MASE and CRPS metrics, along with statistical significance testing, we find that TimesFM and ChronosT5Base consistently outperform classical statistical baselines, while others demonstrate architectural limitations. These findings contribute to a deeper methodological understanding of TSFMs in health-related forecasting scenarios and suggest that performance differences may arise from model design and training data characteristics. The results provide valuable insights to guide future model development and support more informed use of forecasting tools in public health decision-making.

*Index Terms*—Epidemiology, Evaluation models, Public health, Time Series Foundation Models (TSFM).

## I. INTRODUCTION

Effective public health surveillance and response systems rely heavily on accurately forecasting disease trends. These predictions enable proactive resource allocation, such as strategically deploying vaccines and personnel, and implementation of interventions, such as targeted public health campaigns and disease vector control programs [1]. Traditional statistical models, such as Autoregressive Integrated Moving Average (ARIMA), have served this domain well due to their well-established theoretical foundations and interpretability [2]. Although statistical models provide a valuable framework for understanding and forecasting disease patterns, the inherent complexities and evolving nature of epidemiological data often require exploring alternative modeling approaches.

Advancements in surveillance systems have transformed public health, shifting from manual, error-prone data transcription to automated, technology-driven approaches that enhance data quality and reliability. Public health departments are increasingly leveraging modern information and communication technologies to improve disease monitoring [3]. With growing data availability and enhanced computational power, machine learning-based techniques have emerged as valuable tools for disease forecasting, offering new possibilities for more accurate and timely public health surveillance [4]. Building on this foundation, machine learning models have shown promise in addressing some limitations of traditional methods. Their ability to learn complex patterns from data allows them to capture non-linear relationships and improve predictive accuracy [5], [6]. However, these methods face challenges, such as the need for large training datasets and frequent retraining to adapt to evolving disease dynamics, limiting their applicability in resource-constrained public health environments [6]. Furthermore, developing customized models for specific diseases or regions is often impractical because of the requirements of highly specialized expertise and substantial resources.

Time Series Foundation Models (TSFMs), pre-trained on massive and diverse time series datasets, offer a promising solution to these challenges. By leveraging robust architectures like transformers, TSFMs can capture complex temporal dependencies and generalize well to new forecasting tasks without requiring disease-specific training data. Their ability to perform zero-shot forecasting makes them especially attractive for public health applications, as they eliminate the need for training or periodic retraining by specialized data science teams and can integrate seamlessly into existing surveillance software. While TSFMs have shown promising results in domains such as finance and energy [7], their potential for disease forecasting within a public health surveillance context remains underexplored.

This study seeks to fill this gap by systematically assessing the performance of state-of-the-art TSFMs for forecasting notifiable disease trends in Brazil. Our central hypothesis is that TSFMs outperform classical statistical models under zero-shot conditions for epidemiological forecasting. Beyond this core hypothesis, we investigate key research questions to understand the nuances of TSFM performance in this domain:

**RQ1** How does the choice of TSFM architecture influence forecasting performance, particularly when dealing with seasonal patterns and data sparsity?

This study was financed in part by the Coordenação de Aperfeiçoamento de Pessoal de Nível Superior – Brasil (CAPES) – Finance Code 001, and supported by UFOP, CSILab, and the Conselho Nacional de Desenvolvimento Científico e Tecnológico – Brasil (CNPq), under Grant 308400/2022-4.

The source code for the experiments presented in this study is publicly available at: https://github.com/mrlucasrib/TSFM4Epi

**RQ2** How does forecasting precision vary with different forecast horizons and context lengths?

**RQ3** Does the absence of explicit timestamp information in certain TSFMs (e.g. Chronos, TimesFM) limit their ability to capture seasonal dynamics common in public health data?

The investigation of this hypothesis and related research questions is the core of this work, with the aim of clarifying the capabilities and limitations of TSFMs for epidemiological forecasting. We benchmark TSFMs against classical statistical models in both point and probabilistic settings, applying statistical tests to verify significance. Experiments show that TSFMs consistently outperform baselines: TimesFM had the lowest MASE in 9 of 16 diseases, while ChronosT5Base achieved the best CRPS in more than half. These findings highlight TSFMs as accurate, low-maintenance tools for scalable public health forecasting.

## II. BACKGROUND

### A. The Time Series Forecasting Task

Let $Y = \{y_1, y_2, \ldots, y_T\}$ represent a univariate time series, where $y_t \in \mathbb{R}$ is the value observed at discrete time step $t$, and $T$ is the total number of observations available. The fundamental goal of time series forecasting is to predict future values of the series based on its past observations.

Formally, given a history or context window of length $C$, denoted as $Y_{t-C+1:t} = \{y_{t-C+1}, \ldots, y_t\}$, the task is to forecast the next $H$ future values, denoted as the **forecast horizon** $Y_{t+1:t+H} = \{y_{t+1}, \ldots, y_{t+H}\}$. A forecasting model, denoted by $\mathcal{M}$, is a function that maps the context window to a prediction for the forecast horizon:

$$\mathcal{M} : Y_{t-C+1:t} \mapsto \hat{Y}_{t+1:t+H} \tag{1}$$

where $\hat{Y}_{t+1:t+H}$ represents the forecast. The specific nature of this forecast $\hat{Y}_{t+1:t+H}$ distinguishes different approaches.

*a) Point Forecasting:* Traditional methods, such as ARIMA or Exponential Smoothing (ETS), often produce point forecasts. In this case, the output is a single sequence of predicted values:

$$\hat{Y}_{t+1:t+H} = \{\hat{y}_{t+1}, \ldots, \hat{y}_{t+H}\} \tag{2}$$

where each $\hat{y}_{t+h}$ is the single best estimate for the future value $y_{t+h}$. While interpretable, point forecasts do not inherently quantify the uncertainty associated with the prediction.

*b) Probabilistic Forecasting:* In many real-world applications, particularly in public health decision-making, understanding the uncertainty is crucial. Probabilistic forecasting addresses this by predicting a full probability distribution over the future values, rather than a single point estimate [8]. The model $\mathcal{M}$ now outputs a representation of the joint predictive distribution:

$$\mathcal{M} : Y_{t-C+1:t} \mapsto \hat{P}_{t+1:t+H} \tag{3}$$

where $\hat{P}_{t+1:t+H}$ represents the estimated conditional probability distribution $P(Y_{t+1:t+H}|Y_{t-C+1:t})$. In practice, this distribution might be represented parametrically (e.g., by predicting the mean and variance of a Gaussian distribution for each future time step) or non-parametrically (e.g., by predicting specific quantiles of the distribution, allowing for the construction of prediction intervals). This approach allows a richer assessment of future possibilities and associated uncertainties.

### B. Foundation Models for Time Series

Recently, Foundation Models [9] have emerged as a powerful paradigm. These are large-scale models, often based on architectures like the Transformer [10], pre-trained on vast and diverse datasets. Time Series Foundation Models (TSFMs) [7], [11] are foundation models specifically designed or adapted for time series tasks. A key characteristic is their ability to act as the predictor $\mathcal{M}$ in (1) or (3), often in a zero-shot setting (i.e., without any task-specific fine-tuning on the target time series $Y$). They leverage the general patterns and temporal dynamics learned during pre-training to make predictions on unseen series. Many modern TSFMs are designed to produce probabilistic forecasts (3), making them particularly relevant for applications requiring uncertainty quantification.

In this work, we evaluate several state-of-the-art TSFMs, functioning as the predictor $\mathcal{M}$, comparing their zero-shot probabilistic forecasting performance against established statistical models on the task of predicting epidemiological time series.

TABLE I
OVERVIEW OF THE EVALUATED MODELS, THEIR ARCHITECTURES, AND CAPABILITIES FOR TIME SERIES FORECASTING

| Model | Architecture | Timestamps | Prob. Forecasting | Multivariate |
|---|---|---|---|---|
| Chronos | T5-based LM Framework | No | Yes | No |
| Moirai | Encoder | Yes | Yes | Yes |
| Moirai-MoE | Encoder + MoE | Yes | Yes | Yes |
| Lag-Llama | Decoder-only Transformer | Yes | Yes | No |
| Tiny Time Mixers | TSMixer-based | Yes | No | Yes |
| TimesFM | Decoder-only Transformer | No | No | No |

*1) Evaluated Time Series Foundation Models:* This section presents an overview of the specific TSFMs used in this study, highlighting their key features, such as architectural descriptions, timestamp management, support for probabilistic forecasting (as defined by $\hat{P}_{t+1:t+H}$ in (3)), and compatibility with multiple variables. We summarized these characteristics in Table I, consolidating the details discussed in the subsequent paragraphs.

*a) Chronos:* [12] Adapts the T5 architecture, tokenizing data via mean scaling and quantization into bins. It processes tokens as sequences without explicit temporal features, using standard NLP tokens. Observations are modeled via categorical distributions for probabilistic forecasts generated autoregressively. Pre-trained on 890,000 univariate series (84B observations).

*b) TimesFM:* [13] is a decoder-style attention TSFM using input patching (contiguous time point blocks as single input units) for efficiency. Trained on a 100 billion time point corpus from diverse sources (e.g., Wikipedia visits, web searches, synthetic data). It predicts the entire horizon at once, using autoregression if the horizon exceeds its maximum prediction length.

*c) Moirai:* [14] A universal TSFM for any-variate, probabilistic forecasting. It uses multi-patch size projection adaptable to data frequency and Any-variate Attention for multivariate series. Probabilistic outputs use a mixture of parametric distributions (Student's t, negative binomial, log-normal). Pre-trained on LOTSA (27B+ observations), with healthcare (0.01%) and monthly frequency data (0.04%) being minimal in its training.

*d) Moirai-MoE:* [15] An enhanced Moirai with a mixture-of-experts (MoE) architecture. It uses a single projection layer, with sparse specialized experts modeling diverse patterns via token-level specialization, a novel gating function, and a decoder-only training objective.

*e) Lag-Llama:* [16] is a decoder-only transformer model based on the LLaMA architecture. It uses lagged time series features as covariates and incorporates pre-normalization via RMSNorm and Rotary Positional Encoding (RoPE) at each attention layer's query and key representations. It is pre-trained on diverse time series datasets in a univariate fashion. LagLlama was adapted for probabilistic forecasting by using a distribution head at the output and adding value scaling. It was trained with a 32 context length window, but it can optionally process up to 1092 additional historical data points, which are automatically utilized for the construction of its lag-based features.

*f) Tiny Time Mixers (TTM):* [17] employs a lightweight, self-attention-free TSMixer backbone (inspired by MLPMixer) with adaptive patching for varied data resolutions. Training involves pre-training on univariate data then fine-tuning only model heads.

### C. Related Work

Analyzing time series data for epidemiological surveillance has long been a critical tool in public health. For instance, [18] applied an ARIMA model to historical Hemorrhagic Fever with Renal Syndrome (HFRS) incidence data in China, aiming to monitor and forecast short-term trends of this significant public health issue. Their study highlighted the utility of time series models in public health planning. Similarly, [19] conducted a comparative study of four-time series forecasting methods, including ARIMA and Support Vector Machines (SVMs), using surveillance data to predict the incidence of nine infectious diseases in China. Their findings revealed that SVMs generally outperformed other methods, particularly in capturing complex, nonlinear relationships in the data. Additionally, [20] provided an overview of various statistical modeling and prediction techniques for infectious diseases, including the recent COVID-19 pandemic. While these studies demonstrated strong performance in their respective tasks, they are limited by the lack of a generalized model capable of performing well on diverse time series data without requiring task-specific training.

The advent of deep learning models has further advanced the robustness of epidemiological surveillance data analysis. For example, [21] compared three deep learning models to predict the incidence of different types of hepatitis in China using

TABLE II
SUMMARY OF TIME SERIES DATA BY DISEASE

| Code | Disease Name | # TS |
|---|---|---|
| CHIKBR | Chikungunya Fever | 23 |
| DENGBR | Dengue | 27 |
| ESQUBR | Schistosomiasis | 5 |
| EXANBR | Exanthematous Disease | 9 |
| HANSBR | Leprosy | 18 |
| HEPABR | Hepatitis | 15 |
| LEIVBR | Visceral Leishmaniasis | 5 |
| LEPTBR | Leptospirosis | 10 |
| LTANBR | Tegumentary Leishmaniasis | 10 |
| MENIBR | Meningitis | 10 |
| SIFABR | Acquired Syphilis | 22 |
| SIFCBR | Congenital Syphilis | 8 |
| SIFGBR | Gestational Syphilis | 15 |
| TRACBR | Trachoma Survey | 6 |
| VARCBR | Chickenpox | 18 |
| ZIKABR | Zika Virus | 20 |

national surveillance data. Their results showed the robustness of deep learning models in capturing complex patterns. These models can generalize across domains when trained on diverse datasets. This capability could enable non-specialists to predict unseen datasets accurately, even without extensive domain-specific training.

Foundation models, pre-trained on vast and diverse datasets, have shown promise in various forecasting tasks. For instance, [22] compared the performance of time series foundation models, such as TimesFM, to trained-from-scratch Transformer-based models for short-term household electricity load forecasting. Their findings indicated that the TimesFM foundation model outperformed traditional Transformer models, particularly when provided with longer input sequences. Similarly, [23] evaluated the performance of TSFMs in predictive building analytics, finding that these models exhibited marginally better performance compared to statistical models on unseen sensing modalities and patterns. However, their study also highlighted limitations in the generalizability of TSFMs across different datasets. Although these advances are promising, their applicability to epidemiological surveillance data remains an open question. It warrants further exploration to determine whether TSFMs can succeed similarly in public health forecasting tasks.

### III. METHODS

This section details the methodology employed to evaluate the zero-shot forecasting capabilities of the Time Series Foundation Models (TSFMs) on real-world epidemiological data. Our approach centers on an experimental setup designed to compare TSFM performance against established statistical baselines across various conditions. The following subsections describe the data curation process, the specific experimental configurations, the evaluation protocols, and the computational environment used to answer our research questions.

### A. Disease Selection and Categorization

The dataset comprises a curated selection of diseases that align with the specific requirements for the experimental setup.

We categorized the diseases based on their characteristics and prevalence:

*a) Neglected Tropical Diseases:* Neglected Tropical Diseases (NTDs) primarily affect populations in tropical and subtropical regions, where they are often exacerbated by socioeconomic challenges and limited access to healthcare [24]. These diseases include Chikungunya, Dengue, Schistosomiasis, Leprosy, Trachoma, and Visceral and Tegumentary Leishmaniasis [25]. NTDs represent a significant public health burden, with Leishmaniasis being a notable example. In the Americas, nearly 97% of reported cases of Visceral Leishmaniasis occur in Brazil, underscoring the need for targeted interventions in the region to address this critical issue [26].

*b) Infectious Sexually Transmitted Diseases (STDs):* Primarily transmitted sexually, STDs like syphilis (estimated 6 million new annual global cases) require robust surveillance for monitoring, control, and intervention to prevent severe outcomes. [27]

*c) Uncategorized Diseases:* Other relevant diseases include Hepatitis, Meningitis, Chickenpox, Zika Virus, and "Exanthematous Disease" (a cutaneous rash symptom from various pathogens). Monitoring this symptomatic category helps identify broader disease patterns and outbreaks.

### B. Data Acquisition and Processing

*a) Data Source:* The Brazilian Notifiable Diseases Information System (SINAN) provides publicly available, anonymized data on cases of diseases with mandatory notification. This system includes demographic, clinical, and epidemiological variables for reported cases. We utilized this data source to build our experimental datasets.

*b) Data Processing and Aggregation:* The raw SINAN data was processed to create a monthly time series for each state, disease, month, and year by aggregating reported cases. Missing records were filled with zero counts, resulting in a single monthly data point per disease per state, representing the total cases for that location and period.

*c) Disease Selection:* Many diseases in SINAN have low incidence rates or significant underreporting, which resulted in highly sparse time series data. A sufficient amount of data is necessary for models to capture underlying patterns effectively and produce reliable predictions. To address this issue, we selected diseases using the following criteria: 1) $\geq 50$ average monthly cases per state; 2) $\geq 80$ months of series length; and 3) time series from $\geq 5$ states (to ensure experimental variability). This curation yielded a relevant dataset for model evaluation.

### C. Experimental Setup

*a) Model Selection and Setup:* We evaluated a range of state-of-the-art TSFMs, presented in Table I. We used the default parameters for each model based on their documentation. For probabilistic models, 20 samples were used, consistent with the approach used by [12] in their evaluations. For a fair comparison, we evaluated all the models in base-size variants.

*b) Data Input:* We partitioned each time series into non-overlapping windows, where the window size equaled the context length. This ensured sufficient data remained to accommodate the prediction horizon following each context window. As illustrated in Fig. 1, this windowing approach aligns the segments with the specified context and prediction lengths, maintaining consistency across evaluations.

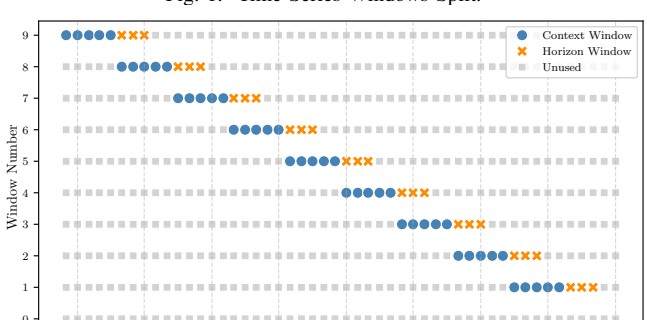

Fig. 1. Time Series Windows Split.

*c) Baselines:* To evaluate the performance of the TSFMs, we selected statistical methods that are widely recognized in the forecasting literature. Following the recommendations of [28], we included benchmark models such as Auto ETS, Auto ARIMA [29], and Auto Theta [30].

### D. Evaluation Protocol

All models used identical input windows for consistent forecasting. Performance was assessed across six configurations: context lengths $C \in \{32, 96, 168\}$ and forecast horizons $H \in \{12 \text{ (short-term)}, 24 \text{ (extended)}\}$ months. This setup allowed analysis of context length effects on cyclical/seasonal series. Experiments were repeated $n = 10$ times, reporting median error for robustness, with results analyzed both aggregately and per configuration.

*a) Evaluation Metrics:* The evaluation metrics used are divided into two categories: point and probabilistic forecasts. For point forecasts, we adhered to the guidelines provided by [31], which underscore common pitfalls in evaluating forecasting models. As recommended, the Mean Absolute Scaled Error (MASE) [31]. MASE calculates the absolute forecast error scaled by the historical seasonal error of the time series, making it particularly suitable for comparison across diverse datasets [31]. Probabilistic forecasting models relied on the median forecast ($0.5-$quantile) to compute the metrics.

For probabilistic forecasts, we adopted a metric commonly used in the literature, Continuous Ranked Probability Score (CRPS) [32].

To compare the performance of the models, we conducted a series of statistical tests. We obtained individual results of MASE for each model across different time series (i.e., States), prediction, and context lengths, leading to a total of 221 paired samples. Following the guidelines provided

Fig. 2. Average ranks and critical distance (CD) with statistical significance $\alpha = 0.05$ according to the Friedman test.

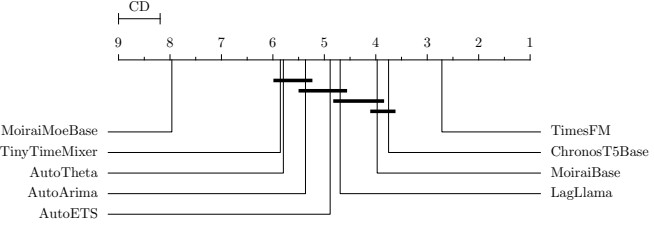

by [31], we applied the Friedman test [33], [34], a non-parametric method suitable for comparing multiple models across several datasets. After identifying significant differences through the Friedman test, we performed post-hoc analysis using the Nemenyi method [35] to pinpoint specific pairwise differences between models.

The significance test results were visualized using Critical Difference (CD) diagrams [36], which graphically represent the ranking of models and the thresholds for significant differences. This comprehensive evaluation framework ensures that the reported results are statistically sound and practically relevant.

### E. Experimental Environment

We conducted the experiments using Python 3.10 within a reproducible environment. Docker containerized the setup, with pip-tools ensuring dependencies were fixed to specific versions. Our experiments were executed on a system featuring an Intel Core i9 processor, an NVIDIA RTX 3090 GPU with 24 GB of VRAM, and 64 GB of RAM.

## IV. RESULTS

### A. Point Forecast Evaluation

Table III shows TimesFM achieved the best performance for most diseases, with ChronosT5Base frequently among the top two. MoiraiBase was second in $\approx 30\%$ of diseases. Conversely, MoiraiMoeBase had consistently higher MASE values. AutoTheta achieved the best performance in three diseases.

*a) Statistical Analysis:* Results show TimesFM as the top-performing model overall. ChronosT5Base and MoiraiBase followed with no statistically significant difference from each other. These TSFMs consistently outperformed traditional statistical methods. LagLlama and TinyTimeMixer did not significantly improve upon baselines. MoiraiMoeBase performed poorest, underscoring TSFMs' general lead over traditional models in this context.

### B. Probabilistic Forecast Evaluation

Table IV shows ChronosT5Base consistently had superior probabilistic accuracy. MoiraiMoeBase again exhibited weaker performance. MoiraiBase ranked second in 50% of the disease cases, indicating a strong level of competitiveness. Auto-Theta showed notable effectiveness in certain syphilis datasets

(SIFABR and SIFCGBR). A particularly important finding is the underperformance of statistical models in the EXANBR dataset, where all TSFMs except MoiraiMoeBase performed well.

*a) Comparative Performance at Different Forecast Horizons and Context Lengths:* Table V shows that TSFMs (excluding MoiraiMoeBase) consistently achieved lower CRPS scores than statistical methods across varying context lengths $C$ and prediction horizons $H$. The performance of statistical models generally worsened as $H$ increased, while most TSFMs maintained stable accuracy. ChronosT5Base benefited from longer context lengths, whereas LagLlama performed better with shorter $C$. Notably, MoiraiMoeBase's performance deteriorated significantly with longer prediction lengths (e.g., CRPS of $3.8937$ at $C = 168, H = 12$)

## V. DISCUSSION

This study validates the central hypothesis that TSFM outperform classical statistical methods in zero-shot epidemiological forecasting. Statistical analysis using the Friedman test confirmed significant performance differences across models, with TimesFM, ChronosT5Base, and MoiraiBase consistently achieving top rankings across diseases and configurations. These results underscore the viability of TSFMs as accurate, low-maintenance forecasting tools for public health surveillance.

*a) **RQ1** Architectural Impact Under Sparsity and Seasonality:* Model architecture strongly influenced forecasting performance under data sparsity and seasonal variation. TimesFM, with its decoder-only transformer and patch-based input representation, demonstrated robust performance across all tasks. ChronosT5Base, despite lacking explicit temporal encodings, performed comparably well, indicating that its tokenization approach with scaling and quantization without changing the Large Language Model (LLM) backbone could capture seasonal patterns without time-stamp features.

In probabilistic forecasting, MoiraiBase was competitive due to its mixture of distributions, which combines three parametric distributions to make probabilistic predictions, a unique feature not employed by any of the other evaluated models. In contrast, MoiraiMoeBase showed poor results across both forecast types. A plausible explanation lies in its sparse Mixture-of-Experts (MoE) architecture: although designed for specialization, its token-level routing may have overfitted to the dominant domains in the LOTSA pretraining corpus, primarily finance and energy, leaving it ill-equipped to model the structure and uncertainty typical of epidemiological data. This suggests that domain-agnostic MoE designs can fail under domain shift, especially when the experts are rarely activated during inference on underrepresented target domains such as health.

Similarly, TinyTimeMixer, designed as a lightweight alternative to Transformer-based models, did not outperform statistical baselines and consistently lagged behind other foundation models. This indicates that the efficiency gains of the TSMixer

### TABLE III
#### MASE METRIC BY DISEASE

| Disease / Model | AutoArima | AutoETS | AutoTheta | ChronosT5Base | LagLlama | MoiraiBase | MoiraiMoeBase | TimesFM | TinyTimeMixer |
|---|---|---|---|---|---|---|---|---|---|
| CHIKBR | 2.9228 | 2.6798 | 2.6802 | *2.6422* | 2.7571 | 2.6612 | 2.9443 | **2.6094** | 2.6917 |
| DENGBR | 2.0517 | 2.0686 | 2.0947 | 1.7930 | 1.7259 | *1.7233* | 2.0129 | **1.6485** | 1.7456 |
| ESQUBR | 0.4959 | 0.4569 | 0.6645 | *0.4236* | 0.5831 | 0.4662 | 1.9508 | **0.4088** | 0.4572 |
| EXANBR | 2.0790 | 1.7957 | 2.0475 | 1.7720 | *1.7653* | 1.8281 | 2.8149 | **1.7277** | 1.8783 |
| HANSBR | 0.9210 | 0.9636 | *0.8537* | 0.8781 | 1.0719 | 0.9414 | 1.7450 | **0.8148** | 1.1571 |
| HEPABR | 1.1396 | 1.0327 | 1.0466 | **0.8625** | 1.0015 | 0.9917 | 1.3727 | *0.9087* | 1.0732 |
| LEIVBR | 1.0738 | 1.1026 | 1.2393 | **0.9895** | 1.0665 | *1.0098* | 1.4478 | 1.0114 | 1.2964 |
| LEPTBR | 1.2032 | 1.1524 | 1.2313 | *0.9627* | 1.1159 | 0.9898 | 1.5490 | **0.9410** | 1.1536 |
| LTANBR | 0.9027 | 0.9877 | 0.9532 | *0.8241* | 0.9063 | 0.8375 | 1.2358 | **0.7789** | 0.9264 |
| MENIBR | 0.8917 | 0.7795 | 0.8808 | **0.6864** | 0.9032 | 0.7862 | 1.6028 | *0.7361* | 0.8725 |
| SIFABR | *1.9610* | 2.1420 | **1.9556** | 2.2355 | 2.6866 | 1.9764 | 4.1237 | 2.3423 | 2.3379 |
| SIFCBR | 1.3821 | 1.3286 | **1.2827** | 1.6145 | 1.5931 | *1.3114* | 2.6021 | 1.4755 | 1.6209 |
| SIFGBR | 1.6682 | 1.6930 | **1.5693** | 2.1252 | 2.1410 | *1.6535* | 4.1428 | 1.8477 | 2.3287 |
| TRACBR | 1.2586 | 1.4552 | 1.3506 | *1.0914* | 1.1243 | 1.1843 | 1.5539 | **1.0910** | 1.2023 |
| VARCBR | 1.0015 | 0.8335 | 0.9841 | **0.6832** | *0.6988* | 0.8429 | 1.8720 | 0.7745 | 0.8219 |
| ZIKABR | 1.0868 | 1.0416 | 1.4720 | 1.0228 | 1.1008 | *0.9736* | 1.5048 | **0.9670** | 1.0314 |

### TABLE IV
#### PROBABILISTIC EVALUATION USING CRPS METRIC BY DISEASE

| Model / Disease | CHIKBR | DENGBR | ESQUBR | EXANBR | HANSBR | HEPABR | LEIVBR | LEPTBR | LTANBR | MENIBR | SIFABR | SIFCBR | SIFGBR | TRACBR | VARCBR | ZIKABR |
|---|---|---|---|---|---|---|---|---|---|---|---|---|---|---|---|---|
| AutoArima | 1.4397 | 1.5871 | 0.8137 | 3.2630 | 0.1352 | 0.2353 | 0.1896 | 0.4624 | 0.2912 | 0.2668 | *0.1920* | 0.1376 | *0.1358* | 0.8071 | 2.3493 | 6.4167 |
| AutoETS | 1.1604 | 1.4907 | 0.7876 | 4.6159 | 0.1440 | 0.2012 | 0.1864 | 0.5592 | 0.3419 | 0.2489 | 0.2023 | 0.1320 | 0.1374 | 0.9873 | 3.5666 | 2.3861 |
| AutoTheta | 1.1322 | 1.4052 | 0.8282 | 5.0568 | *0.1265* | 0.2054 | 0.2152 | 0.5063 | 0.3104 | 0.2794 | 0.1952 | *0.1270* | *0.1291* | 1.0094 | 2.3455 | 2.6457 |
| ChronosT5Base | *0.8169* | 1.1169 | **0.2441** | **0.6043** | *0.1293* | **0.1754** | 0.1704 | *0.3369* | *0.2613* | 0.1721 | 0.2111 | 0.1704 | 0.1877 | 0.7919 | *0.7261* | *0.6393* |
| LagLlama | **0.8050** | **0.8600** | 0.4652 | *0.7415* | 0.1731 | 0.2041 | 0.1849 | 0.3913 | 0.2863 | 0.2800 | 0.2340 | 0.1669 | 0.1860 | **0.7203** | 1.5158 | 0.6866 |
| MoiraiBase | 0.8368 | *0.9562* | *0.2740* | 0.7600 | 0.1343 | *0.1897* | *0.1719* | **0.3335** | **0.2473** | *0.2010* | 0.1779 | *0.1311* | 0.1361 | *0.7307* | *1.2872* | **0.6310** |
| MoiraiMoeBase | 1.3178 | 1.1853 | 2.8788 | 5.3240 | 0.3031 | 0.3394 | 0.2349 | 0.7241 | 0.4466 | 0.6663 | 0.3976 | 0.2520 | 0.3699 | 1.0489 | 8.1877 | 3.4034 |

### TABLE V
#### CRPS FOR EACH MODEL WITH CONTEXT AND HORIZON

| Model | H/C | 32 | 96 | 168 |
|---|---|---|---|---|
| AutoArima | 12 | 0.8490 | 0.5482 | 1.2887 |
| AutoETS | 12 | 0.6719 | 0.6420 | 1.7704 |
| AutoTheta | 12 | 0.7044 | 0.6160 | 1.5346 |
| ChronosT5Base | 12 | 0.4232 | 0.3552 | 0.3120 |
| LagLlama | 12 | 0.4020 | 0.3715 | 0.6664 |
| MoiraiBase | 12 | 0.4215 | 0.3678 | 0.4097 |
| MoiraiMoeBase | 12 | 0.8728 | 0.9375 | 3.8937 |
| AutoArima | 24 | 1.3269 | 0.6040 | 1.1858 |
| AutoETS | 24 | 0.8359 | 0.7084 | 1.9183 |
| AutoTheta | 24 | 0.9076 | 0.6684 | 1.6979 |
| ChronosT5Base | 24 | 0.4576 | 0.4717 | 0.3407 |
| LagLlama | 24 | 0.4457 | 0.4219 | 0.6398 |
| MoiraiBase | 24 | 0.4538 | 0.4653 | 0.4800 |
| MoiraiMoeBase | 24 | 0.9264 | 1.0238 | 3.2004 |

architecture may not be sufficient to offset the representational power of Transformer-based models in this domain.

*b) **RQ2** Forecast Horizon and Context Length Effects:* Across different prediction horizons and context lengths, TSFMs consistently outperformed traditional statistical baselines. In particular, ChronosT5Base showed improved performance as the input sequence length increased. This can be attributed to the way the model handles shorter sequences, where padding tokens are used when the input is insufficient. These tokens carry no temporal information, and as more actual data becomes available, the model can better learn the underlying dynamics of the series.

In contrast, LagLlama performed best with shorter context windows. This probably reflects its pre-training setup, which used sequences of 32 timepoints. In LagLlama, longer contexts are treated as lagged features based on fixed offsets rather than contiguous inputs. We hypothesize that this structure, especially when applied to longer sequences, may introduce misleading dependencies. As a result, the model could misinterpret temporal relationships, leading to reduced performance.

*c) **RQ3** Timestamp Information and Seasonality:* Despite omitting explicit timestamp inputs, both TimesFM and ChronosT5Base successfully modeled seasonal epidemiological trends, likely due to their training on extensive and diverse corpora. Their strong performance suggests that pretraining on diverse temporal sequences and their architectural choices can compensate for the absence of calendar-based features. Meanwhile, LagLlama, which includes timestamp encodings and also is based on decoder-only Transformer, did not surpass these models, indicating that timestamp-aware design alone does not guarantee better seasonality handling. These findings suggest that rich temporal representations can emerge from data-driven embeddings when supported by scalable architectures and large training corpora.

### A. Limitations and Future Work

This study is limited to univariate time series, as half of the evaluated models, including leading approaches such as TimesFM and ChronosT5Base, do not support multivariate inputs. Incorporating multivariate data can improve the modeling of disease dynamics and remains a promising avenue for future research.

The datasets employed are restricted to monthly resolution and originate exclusively from SINAN, focusing the analysis on Brazil. Although higher-frequency data could offer finer temporal detail, sparse reporting for many diseases renders this approach impractical. While this geographic focus may

affect generalizability, Brazil's substantial burden of neglected tropical diseases makes it a particularly relevant context for this investigation [25].

Future work should consider more diverse epidemiological datasets, support for multivariate modeling, and fine-tuning strategies. Additionally, analyses of internal model behavior, such as expert activation patterns in Mixture-of-Experts architectures, may yield deeper insights into model performance under domain-specific conditions.

## VI. Conclusion

This study systematically evaluates the zero-shot forecasting capabilities of state-of-the-art Time Series Foundation Models (TSFMs) for notifiable disease trends, testing the hypothesis that TSFMs outperform classical statistical models without disease-specific training. Guided by three research questions, we benchmark multiple TSFMs across real-world epidemiological datasets from Brazil, varying forecast horizons and context lengths. Using robust metrics (MASE, CRPS) and statistical tests, our findings show that models like TimesFM and ChronosT5Base consistently deliver superior accuracy, while others, such as MoiraiMoeBase, exhibit architectural limitations. These findings contribute to a deeper understanding of how TSFMs behave when applied to epidemiological data and raise hypotheses about how model performance may be influenced by architectural design choices and the characteristics of training data, offering valuable insights for the development and deployment of foundation models in health-related forecasting scenarios.

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
