# OpenReview forum: "How Effective Are Time Series Foundation Models for Epidemiological Data Analysis?"
_IEEE.org/EMBS/BHI/2025/Conference — BHI 2025_

### Official Review · Reviewer_oxye · 2025-06-25

**Confidence:** 4
**Clarity Of Writing:** excellent
**Clinical Significance:** good
**Methodological Novelty:** good
**Overall Rating:** 6

**Experiments And Results:**

great

**Questions For The Authors:**

1. Can you provide empirical evidence (e.g., by analyzing expert activation frequencies) to support the hypothesis that Moirai-MoE's failure is due to a domain shift from its pre-training data?
2. What specific architectural differences explain why ChronosT5Base improves with longer context windows while LagLlama's performance degrades?
3. How might a "few-shot" fine-tuning experiment change the performance landscape? Could minimal fine-tuning rescue an underperforming model like Moirai-MoE, suggesting its issue is one of adaptation rather than a fundamental architectural flaw?

**Strengths:**

1. A comprehensive model comparison, use of real-world data, robust evaluation metrics (MASE, CRPS), and appropriate statistical significance testing.
2. Provide insights on the observed performance differences between models, including the valuable negative result of Moirai-MoE's consistent underperformance.

**Summary Of The Paper:**

This paper systematically evaluates the zero-shot forecasting performance of state-of-the-art Time Series Foundation Models (TSFMs) on real-world epidemiological data from Brazil. The study benchmarks models like TimesFM and ChronosT5Base against classical statistical baselines (e.g., AutoARIMA) across 16 diseases, using MASE and CRPS metrics. The central finding confirms the hypothesis that TSFMs outperform traditional methods, with TimesFM and ChronosT5Base showing the strongest performance for point and probabilistic forecasting, respectively.

**Weaknesses:**

1. The study lacks a comparison to a standard deep learning model (e.g., LSTM) trained from scratch. This makes it difficult to distinguish the benefits of the TSFMs' pre-training from the advantages of their underlying architecture.
2. The findings are based on univariate, monthly data from a single country (Brazil), which may not apply to different geographical regions, data frequencies, or multivariate scenarios.
3. The paper presents compelling but unverified hypotheses for key results, such as the failure of the Moirai-MoE model and the opposite effects of context length on LagLlama and ChronosT5Base.

---

### Official Review · Reviewer_cHCn · 2025-07-03
**A rigorous evaluation of zero-shot TSFMs in epidemiology**

**Confidence:** 5
**Clarity Of Writing:** excellent
**Clinical Significance:** excellent
**Methodological Novelty:** good
**Overall Rating:** 8

**Experiments And Results:**

excellent

**Questions For The Authors:**

The study demonstrates the power of TSFMs on monthly data, but public health officials often need higher frequency forecasts (e.g., weekly) for timely interventions. How do you hypothesize the performance of models like TimesFM and ChronosT5Base would change when faced with higher-frequency but consequently more sparse and noisy epidemiological data?

**Strengths:**

The study features a good experimental design, comparing several Time Series Foundation Models (TSFMs) to statistical baselines using standard metrics like MASE and CRPS. Results are validated with robust statistical tests, ensuring reliable conclusions.
Its relevance is clear. It uses real epidemiological data from Brazil, and the research addresses public health forecasting needs. The focus on zero-shot forecasting makes it especially useful for settings with limited technical resources.
Lastly, the authors analyze why some models perform better, linking results to architectural choices. For instance, they suggest MoiraiMoeBase struggled due to its pretraining on financial data, offering valuable insights for future model design.

**Summary Of The Paper:**

This study examines the zero-shot forecasting performance of Time Series Foundation Models (TSFMs) using monthly epidemiological data from Brazil’s SINAN system, covering 16 diseases such as Dengue and Syphilis. The goal was to test whether TSFMs could outperform traditional models like Auto ARIMA, ETS, and Theta without task-specific training.
Several TSFMs, including TimesFM, ChronosT5Base, and MoiraiBase, were benchmarked using point (MASE) and probabilistic (CRPS) forecasting metrics. The models were tested across varying time windows and forecast horizons. Results showed that TimesFM and ChronosT5Base outperformed classical models. TimesFM had the best point forecast accuracy for over half the diseases, while ChronosT5Base led in probabilistic accuracy.
Model architecture played an important role in performance. Moirai-MoE, designed for finance and energy data, underperformed on epidemiological patterns. In contrast, top models performed well even without explicit timestamp inputs, suggesting that large-scale pretraining helps capture temporal trends.
In summary, TSFMs, especially TimesFM and ChronosT5Base, are promising tools for scalable and low-maintenance public health forecasting. However, limitations include the use of only univariate data and focus on Brazilian datasets.

**Weaknesses:**

The study is limited by its geographic scope, relying solely on data from Brazil, which may reduce the applicability of its findings to other regions with different health systems and disease dynamics. Additionally, the analysis is restricted to univariate forecasting, using only past case counts without considering external factors such as climate or demographics that could influence disease trends. Finally, the use of monthly data avoids issues with sparsity but limits the ability to assess model performance for short-term forecasts, which are often important for time-sensitive public health interventions.

---

### Official Review · Reviewer_jpWu · 2025-07-14
**How Effective Are Time Series Foundation Models for Epidemiological Data Analysis?**

**Confidence:** 4
**Clarity Of Writing:** great
**Clinical Significance:** good
**Methodological Novelty:** great
**Overall Rating:** 7

**Experiments And Results:**

great

**Questions For The Authors:**

- Have you considered benchmarking “large” or “small” variants of these models to understand trade-offs between parameter count, pretraining corpus size, and epidemiological performance?
- Can you provide more analysis to explain why the MoE design fails under domain shift to health data?
- Can you release the exact Dockerfile and random seeds used, to ensure end-to-end reproducibility across different compute environments?

**Strengths:**

- comprehensive zero-shot assessment of TSFMs in a public health surveillance context
- Evaluates a diverse set of modern architectures under multiple context-horizon configurations and repeats experiments (n=10) to report median performance.
- Applies non-parametric Friedman tests and post-hoc Nemenyi analyses, accompanied by critical-difference diagrams to support claims of model superiority.
- Experiments are containerized via Docker, and code is publicly available

**Summary Of The Paper:**

This study conducts a zero-shot evaluation of several classes of pre-trained time-series models against classical statistical approaches for forecasting monthly disease incidence. By systematically varying the length of historical data used and the forecast horizon, and assessing both point and probabilistic accuracy with robust error metrics and non-parametric significance tests, the authors demonstrate that modern foundation models consistently outperform traditional baselines in most scenarios. A subset of architectures, however, fails to surpass the statistical methods, highlighting that not all pre-trained designs generalize equally to epidemiological data. These findings underscore the promise of foundation models as low-maintenance, accurate tools for public-health forecasting while revealing how architectural choices and pretraining corpora influence performance.

**Weaknesses:**

- Only “base-size” variants are benchmarked; it remains unclear whether larger or smaller model variants would alter performance rankings.
- The paper does not report inference latency or resource requirements.
- Focuses exclusively on monthly Brazilian data; findings may not generalize to higher-frequency series

---

### Official Review · Reviewer_n6sQ · 2025-07-18
**A well-structured evaluation of pre-trained models for disease forecasting with strong practical implications.**

**Confidence:** 5
**Clarity Of Writing:** great
**Clinical Significance:** good
**Methodological Novelty:** great
**Overall Rating:** 7

**Experiments And Results:**

great

**Questions For The Authors:**

Could you clarify whether any of the TSFMs were exposed to public health or epidemiological datasets during pretraining? This would help assess whether the observed performance reflects generalization or prior domain exposure, which could influence how broadly the findings apply.

For Moirai-MoE, is there any indication from the attention patterns or expert activation behavior that explains the model’s underperformance on these datasets? A brief analysis might help distinguish between architectural mismatch and domain sparsity, which could change how its results are interpreted.

**Strengths:**

The study addresses an important public health challenge by evaluating scalable forecasting tools for disease surveillance using real-world data.

It offers a comprehensive comparison across six foundation models and classical baselines, using consistent, well-documented experimental settings.

The zero-shot design reflects practical deployment conditions, where retraining on specific diseases may not be feasible.

TimesFM and ChronosT5 show strong, consistent performance across multiple diseases and forecasting horizons, underscoring their potential utility in low-resource settings.

The authors provide detailed insights into how model architecture, pretraining corpus, and input design influence performance—adding depth to the analysis beyond raw accuracy metrics.

The dataset is carefully curated to ensure quality and relevance, covering diverse diseases with varying seasonality and case volume.

**Summary Of The Paper:**

This paper examines how well large, pre-trained time series foundation models perform in forecasting infectious disease trends, without any additional training on the target data. Using monthly case counts from Brazil’s national health database, the authors evaluate six foundation models—including TimesFM and ChronosT5—alongside standard statistical baselines like ARIMA and ETS. The models are tested across 16 diseases with varying seasonal patterns and sparsity, using different forecast horizons and historical context lengths. Performance is measured using standard metrics for point and probabilistic forecasts.

The study finds that foundation models, especially TimesFM and ChronosT5, generally outperform traditional methods across most settings. These models were able to capture seasonal trends and make accurate predictions even without using explicit date or time inputs. The authors highlight that model architecture and pretraining data are key to success, and suggest that these approaches could be integrated into public health systems to support disease forecasting with minimal tuning or manual effort.







Ask ChatGPT

**Weaknesses:**

The models are evaluated exclusively on univariate time series; many public health forecasting tasks benefit from multivariate inputs (e.g., climate, mobility, policy changes). Extending the analysis to multivariate forecasting could help assess broader applicability.

Some foundation models (e.g., Moirai-MoE, TinyTimeMixer) underperform consistently, but the paper does not fully explore whether this is due to architecture design, domain mismatch, or data sparsity. A deeper investigation could clarify whether these limitations are model-specific or dataset-related.

While the study demonstrates strong performance in a zero-shot setting, the absence of any fine-tuned baseline comparisons limits the reader’s ability to judge how much improvement pretraining alone contributes relative to domain adaptation.